# A New Topological Approach to Target the Existence of Solutions for Nonlinear Fractional Impulsive Wave Equations

Svetlin G. Georgiev [1], Keltoum Bouhali [2,3] and Khaled Zennir [2,4,*]

1 Department of Differential Equations, Faculty of Mathematics and Informatics, University of Sofia, 1504 Sofia, Bulgaria
2 Department of Mathematics, College of Sciences and Arts, Qassim University, Ar-Rass 51452, Saudi Arabia
3 Department of Mathematics, Faculty of Sciences, 20 Aout 1955 University, Skikda 21000, Algeria
4 Laboratoire de Mathématiques Appliquées et de Modélisation, Université 8 Mai 1945 Guelma, Guelma 24000, Algeria
* Correspondence: k.zennir@qu.edu.sa

**Abstract:** This paper considers a class of fractional impulsive wave equations and improves a previous results. In fact, this paper adopts a new topological approach to prove the existence of classical solutions with a complex arguments caused by impulsive perturbations. To the best of our knowledge, there is a severe lack of results related to such impulsive equations.

**Keywords:** fractional impulsive wave equations; classical solutions; fixed point; cone; sum of operators





## 1. Introduction

The theory of nonlinear waves is still a young sciences, although research in this direction was carried out even in the 19th century, mainly in connection with the problems of gas and hydrodynamics. For example, the works of J. Scott Russell [1] who was the first to observe solutions on the surfaces of a liquid, date back to 1830–1840. Nonlinear wave pgenomena have been the subject of research by such outstanding scientists as Poison, Stokes, Airy, Rayleigh, Boussinesq and Riemann. However, as a unified science, the theory of nonlinear waves developed in the late 1960s and early 1970s, which were the years of its rapid development.

This type of problem appears in several mathematical models which describe wave phenomena in areas such as fluid dynamics and electromagnetism. Many authors such as H. Brésiz, J. Mawhin, K. C. Chang and others, have developed topological tools, index theory and variational methods to obtain a classical existence results for the one-dimensional problem with various non-linearities. One can review the associated results in [2–5] and the references therein.

A fractional derivative is a non-local characteristic of a function: it depends not only on the behavior of the function in the vicinity of the point $x$ under consideration, but also on the values it takes over the entire interval $(a, x)$. This non-locality means that the change in the particle flux density depends not only on its values in the vicinity of the point under consideration, but also on its values at distant points in space. We mention some related results on the impulsive equation in [6–10] and these models have not been sufficiently studies, despite their versability and practical importance.

To begin with, let $x \in \mathbb{R}^n, n_1 \in \mathbb{N}, k = 1, \ldots, n_1$. Let $v = v(x, t)$, we consider the following problem

$$
\begin{aligned}
{}^c D^{\beta}_{t,0+} v - \Delta_x v &= f(t, x, v, \partial_t v, v_x), \quad t \in J = [0, 1], \quad t \neq t_k, \\
\partial_t v(x, t_k+) &= \partial_t v(t_k-, x) + I_k(x, t_k, v(x, t_k)), \\
v(x, t_k+) &= v(t_k-, x) + L_k(x, t_k, v(x, t_k)), \\
v(x, 0) &= h_1(x, v(x, 0)), \quad v(x, 1) = h_2(x, v(x, 1)),
\end{aligned}
\tag{1}
$$

where $f, h_1, h_2, l_k, L_k, k \in \{0, ., n_1\}$ satisfy the conditions (Hyp1)-(Hyp4) stated in the next section.

Our aim is to investigate the problem (1) for existence and nonuniqueness of classical solutions. To prove our main results, firstly we reduce the problem (1) to suitable integral equation. Then, we define two operators so that any fixed point of their sum is a solution of the problem (1). In the end, we use some recent fixed point theorems to prove that the sum of the defined two operators has at least one and at least two fixed points in suitable defined spaces. To our knowledge, there is no any research on existence of solutions for the problem (1).

The paper is organized as follows. In Section 2, we give the main assumptions and we state the main results in the paper. In Section 3, we give some preliminary results needed for the proof of our main results. In Section 4, we prove existence of at least one classical solution for the problem (1). In Section 5, we prove existence of at least two classical solutions for the problem (1). In Section 6, we give an illustrative example. A conclusion is provided in Section 7.

## 2. Main Results

**(Hyp1)** ${}^c D^{\beta}_{t,0+}$ is the Caputo fractional derivative with respect to $t$, $\beta \in (1, 2]$, $0 = t_0 < t_1 < \ldots < t_{n_1} < t_{n_1+1} = 1$, $J_0 = [0, t_1]$, $J_1 = (t_1, t_2]$, …, $J_{n_1} = (t_{n_1}, 1]$.

**(Hyp2)** $I_k, L_k \in \mathcal{C}([0, T] \times \mathbb{R}^{n+1})$,

$$
|I_k(t_k, x, v(x, t_k))| \leq a_{1k}(x, t_k)|v(x, t_k)|^{s_{1k}},
$$

$$
|L_k(x, t_k, v(x, t_k))| \leq a_{2k}(x, t_k)|v(x, t_k)|^{s_{2k}},
$$

$a_{1k}, a_{2k} \in \mathcal{C}(J \times \mathbb{R}^n), 0 \leq a_{1k}, a_{2k} \leq \mathcal{B}$ on $J \times \mathbb{R}^n$, for some positive constant $\mathcal{B}$, $s_{1k}, s_{2k} \geq 0$.

**(Hyp3)** $h_1, h_2 \in \mathcal{C}^2(\mathbb{R}^{n+1})$,

$$
|h_1(x, v(x, 0))| \leq b_{11}(x)|v(x, 0)|^{s_1},
$$

$$
|h_2(x, v(x, 1))| \leq b_{12}(x)|v(x, 1)|^{s_2},
$$

$b_{11}, b_{12} \in \mathcal{C}(\mathbb{R}^n), 0 \leq b_{11}, b_{12} \leq \mathcal{B}$ on $\mathbb{R}^n, s_1, s_2 \geq 0$.

**(Hyp4)** $f \in \mathcal{C}(J \times \mathbb{R}^n \times \mathbb{R} \times \mathbb{R} \times \mathbb{R}^n)$,

$$
|f(t, x, v, u, w)| \leq \sum_{j=1}^{r} \left( a_j(x, t)|v|^{p_j} + b_j(x, t)|u|^{q_j} + \sum_{i=1}^{n} c_{ji}(x, t)|w_i|^{r_{ji}} \right),
$$

$(x, t) \in J \times \mathbb{R}^n, v, u \in \mathbb{R}, w \in \mathbb{R}^n, a_j, b_j, c_{ji} \in \mathcal{C}(J \times \mathbb{R}^n), \mathcal{B} \leq a_j, b_j, c_{ji} \leq 0$ on $J \times \mathbb{R}^n, p_j, r_{ji}, q_j > 0, i \in \{1, \ldots, n\}, j \in \{1, \ldots, r\}, r \in \mathbb{N}$.

Here $v_x = (v_{x_1}, \ldots, v_{x_n})$, $\partial_t v(t_k-, x) = \lim\limits_{t \to t_k-} \partial_t v$, $\partial_t v(x, t_k+) = \lim\limits_{t \to t_k+} \partial_t v$, $v(t_k-, x) = \lim\limits_{t \to t_k-} v$, $v(x, t_k+) = \lim\limits_{t \to t_k+} v$. For $l, s \in \mathbb{N} \cup \{0\}$, define

$$
\begin{aligned}
PC(J) \;=\;& PC^0(J) \\[2mm]
=\;& \{g : J \to \mathbb{R}, \quad g \in \mathcal{C}(J \backslash \{t_j\}_{j=1}^{m-1}), \\[2mm]
& \exists g(t_j+), \quad g(t_j-) \quad \text{and} \quad g(t_j-) = g(t_j), \\[2mm]
& j \in \{1, \ldots, n_1\}\}, \\[2mm]
PC^l(J) \;=\;& \{g : J \to \mathbb{R}, \quad g \in PC^{l-1}(J), \quad g \in \mathcal{C}^l(J \backslash \{t_j\}_{j=1}^{n_1}), \\[2mm]
& \exists g^{(l)}(t_j-), g^{(l)}(t_j+) \quad \text{and} \quad g^{(l)}(t_j-) = g^{(l)}(t_j), \\[2mm]
& j \in \{1, \ldots, n_1\}\},
\end{aligned}
$$

and

$$
\begin{aligned}
PC^l(J, \mathcal{C}^s(\mathbb{R}^n)) \;=\;& \{v : v(\cdot, x) \in PC^l(J), \\[2mm]
& v(t, \cdot) \in \mathcal{C}^s(\mathbb{R}^n), \quad t \in J\}.
\end{aligned}
$$

In $PC^2(J, \mathcal{C}^2(\mathbb{R}^n))$, we define the norm

$$
\begin{aligned}
\|v\| \;=\; \max\Big\{ & \max_{j \in \{0,1,\ldots,n_1\}} \sup_{(x,t) \in [t_j, t_{j+1}] \times \mathbb{R}^n} |v|, \\[2mm]
& \max_{j \in \{0,1,\ldots,n_1\}} \sup_{(x,t) \in [t_j, t_{j+1}] \times \mathbb{R}^n} |\partial_t v|, \\[2mm]
& \max_{j \in \{0,1,\ldots,n_1\}} \sup_{(x,t) \in [t_j, t_{j+1}] \times \mathbb{R}^n} |\partial_{tt} v|, \\[2mm]
& \max_{j \in \{0,1,\ldots,n_1\}} \sup_{(x,t) \in [t_j, t_{j+1}] \times \mathbb{R}^n} |v_{x_i}|, \\[2mm]
& \max_{j \in \{0,1,\ldots,n_1\}} \sup_{(x,t) \in [t_j, t_{j+1}] \times \mathbb{R}^n} |v_{x_i x_i}|, \quad i \in \{1, \ldots, n\}\Big\},
\end{aligned}
$$

as long as it exists. Here $PC^2(J, \mathcal{C}^2(\mathbb{R}^n))$ is a Banach space.

We are now in position to state the main results.

**Theorem 1.** *Let $(Hyp1)$–$(Hyp4)$ hold. Then, the problem (1) has a solution in $PC^2(J, \mathcal{C}^2(\mathbb{R}^n))$.*

**Theorem 2.** *Let $(Hyp1)$–$(Hyp4)$ hold. Then, the problem (1) has at least two solutions in $PC^2(J, \mathcal{C}^2(\mathbb{R}^n))$.*

## 3. Preliminary

Here, as in [6], we introduce some preliminary tools and results which will be used to prove our main results. The fixed point theorem for sum of two operators will be used to prove the existence of at least one solution to the problem (1).

**Theorem 3.** *Let $\mathcal{E}$ be a Banach space. For $\epsilon \in (0,1)$ and $0 < \mathcal{B}$ we define*

$$\mathcal{X} = \{x \in \mathcal{E} : \|x\| \leq \mathcal{B}\}.$$

*Let $Tx = -\epsilon x, x \in \mathcal{X}$, and $S : \mathcal{X} \to \mathcal{E}$ is continuous, $(I - S)(\mathcal{X})$ resides in a compact subset of $\mathcal{E}$ and*

$$\{x \in \mathcal{E} : x = \lambda(I - S)x, \quad \|x\| = \mathcal{B}\} = \varnothing, \qquad \forall \lambda \in \left(0, \frac{1}{\epsilon}\right). \tag{2}$$

*Then, there exists a $x^* \in \mathcal{X}$ so that*

$$Tx^* + Sx^* = x^*.$$

*Here $\mu\mathcal{X} = \{\mu x : x \in \mathcal{X}\}, \qquad \forall \mu \in \mathbb{R}.$*

**Proof.** Define

$$r\left(-\frac{1}{\epsilon}x\right) = \begin{cases} -\frac{1}{\epsilon}x & \text{if} \quad \|x\| \leq \mathcal{B}\epsilon \\[2mm] \frac{\mathcal{B}x}{\|x\|} & \text{if} \quad \|x\| > \mathcal{B}\epsilon. \end{cases}$$

Then, $r\left(-\frac{1}{\epsilon}(I - S)\right) : \mathcal{X} \to \mathcal{X}$ is continuous and compact. Then, owing to the Schauder fixed point theorem, there exists $x^* \in \mathcal{X}$ such that

$$r\left(-\frac{1}{\epsilon}(I - S)x^*\right) = x^*,$$

where $-\frac{1}{\epsilon}(I - S)x^* \notin \mathcal{X}$. Thus

$$\left\|(I - S)x^*\right\| > \mathcal{B}\epsilon, \qquad \frac{\mathcal{B}}{\|(I - S)x^*\|} < \frac{1}{\epsilon},$$

and

$$x^* = \frac{\mathcal{B}}{\|(I - S)x^*\|}(I - S)x^* = r\left(-\frac{1}{\epsilon}(I - S)x^*\right),$$

and hence, $\|x^*\| = \mathcal{B}$. This contradicts with (2). Therefore, $-\frac{1}{\epsilon}(I - S)x^* \in \mathcal{X}$ and

$$x^* = r\left(-\frac{1}{\epsilon}(I - S)x^*\right) = -\frac{1}{\epsilon}(I - S)x^*,$$

or

$$-\epsilon x^* + Sx^* = x^*,$$

or

$$Tx^* + Sx^* = x^*.$$

The proof is now completed. $\square$

Let $\mathcal{X}$ be a real Banach space.

**Definition 1.** *We say that a mapping $K : \mathcal{X} \to \mathcal{X}$ is completely continuous, if $K$ is continuous and maps bounded sets into relatively compact sets.*

The concept of contraction of the set $l$ is linked to that of the Kuratowski measure of non-compactness which we recall for completeness.

**Definition 2.** *Let $\Omega_{\mathcal{X}}$ be the class of all bounded sets of $\mathcal{X}$. The Kuratowski measure of noncompactness $\alpha : \Omega_{\mathcal{X}} \to [0, \infty)$ is defined, for $j = 1, \dots, m$, by*

$$\alpha(\varpi) = \inf \left\{ \delta_x > 0 : \varpi = \bigcup_{j=1}^{m} \varpi_j, diam(\varpi_j) \leq \delta_x \right\},$$

*where $diam(\varpi_j) = \sup\{\|x - y\|_{\mathcal{X}} : x, y \in \varpi_j\}$ is the diameter of $\varpi_j$.*

For more related detail on the properties for measure of noncompactness, we refer to [11].

**Definition 3** ([12])**.** *We say that the mapping $\mathcal{K} : \mathcal{X} \to \mathcal{X}$ is l-set contraction, if $\mathcal{K}$ is continuous, bounded and there exists a positive constant $l \geq 0$ s. t.*

$$\alpha(\mathcal{K}(\varpi)) \leq l\alpha(\varpi),$$

*for all bounded set $\varpi \subset \mathcal{X}$. We say that he mapping $\mathcal{K}$ is strict set contraction if $l < 1$.*

**Remark 1.** *If $\mathcal{K} : \mathcal{X} \to \mathcal{X}$ is a completely continuous mapping, then it is 0-set contraction (see [13]).*

**Definition 4.** *Let $\mathcal{X}$ and $\varpi$ be real Banach spaces. We say that the mapping $\mathcal{K} : \mathcal{X} \to \varpi$ is expansive if there exists a constant $h > 1$ such that*

$$\|\mathcal{K}x - \mathcal{K}y\|_{\varpi} \geq h\|x - y\|_{\mathcal{X}}, \qquad \forall x, y \in \mathcal{X}.$$

**Definition 5.** *We say that the closed, convex set $\mathcal{P}$ in $\mathcal{X}$ is cone if*

1. $\alpha x \in \mathcal{P}, \qquad \forall \alpha \geq 0, \qquad \forall x \in \mathcal{P},$
2. $x, -x \in \mathcal{P}$ *implies* $x = 0$.

Denote $\mathcal{P}^* = \mathcal{P} \backslash \{0\}$.

**Lemma 1** ([12])**.** *Let $\mathcal{X}$ be a closed convex subset of a Banach space $\mathcal{E}$ and $v \subset \mathcal{X}$ a bounded open subset with $0 \in \mathcal{U}$. For $0 < \varepsilon$ small enough, we assume that $\mathcal{K} : \overline{\mathcal{U}} \to \mathcal{X}$ is a strict k-set contraction satisfying*

$$\mathcal{K}x \notin \{x, \lambda x\}, \forall x \in \partial\mathcal{U} \text{ and } \lambda \geq 1 + \varepsilon.$$

*Then,*

$$i(\mathcal{K}, \mathcal{U}, \mathcal{X}) = 1.$$

**Proof.** We consider the homotopic functional $\mathcal{H} : [0, 1] \times \overline{\mathcal{U}} \to \mathcal{X}$ given by

$$\mathcal{H}(t, x) = \frac{t\mathcal{K}x}{\varepsilon + 1}.$$

The operator $\mathcal{H}$ is continuous and uniformly continuous in $t$ for each $x$, and the mapping $\mathcal{H}(t, .)$ is a strict set contraction for each $0 \leq t \leq 1$. In addition, $\mathcal{H}(t, .)$ has no fixed point on $\partial\mathcal{U}$. On the other hand:

- If $t = 0$, there exists some $x_0 \in \partial\mathcal{U}$ such that $x_0 = 0$, contradicting $x_0 \in \mathcal{U}$.
- If $t \in (0, 1]$, there exists some $x_0 \in \mathcal{P} \cap \partial\mathcal{U}$ such that $\frac{1}{\varepsilon+1}t\mathcal{K}x_0 = x_0$; then, $\mathcal{K}x_0 = \frac{1+\varepsilon}{t}x_0$ with $\frac{1+\varepsilon}{t} \geq 1 + \varepsilon$, contradicting our assumption. From the invariance under homotopy and the normalization properties of the index, we have

$$i(\frac{1}{\varepsilon+1}\mathcal{K}, \mathcal{U}, \mathcal{X}) = i(0, \mathcal{U}, \mathcal{X}) = 1.$$

Now, we have to prove that

$$i\left(\mathcal{K}, \mathcal{U}, \mathcal{X}\right) = i\left(\frac{1}{\varepsilon+1}\mathcal{K}, \mathcal{U}, \mathcal{X}\right).$$

We have

$$\frac{1}{\varepsilon+1}\mathcal{K}x \neq x, \ \forall\, x \in \partial\mathcal{U}. \tag{3}$$

Then, there exists a positive conatant $\gamma$ such that

$$\gamma \leq \|x - \frac{1}{\varepsilon+1}\mathcal{K}x\|, \ \forall\, x \in \partial\mathcal{U}.$$

In other side, we have $\frac{1}{\varepsilon+1}\mathcal{K}x \to \mathcal{K}x$ as $\epsilon \to 0$, for $x \in \overline{\mathcal{U}}$. So for $\varepsilon$ small enough

$$\frac{\gamma}{2} > \|\mathcal{K}x - \frac{1}{\varepsilon+1}\mathcal{K}x\|, \ \forall\, x \in \partial\mathcal{U}.$$

Let us now define a convex functional $\mathcal{G} : [0,1] \times \overline{\mathcal{U}} \to \mathcal{X}$ by

$$\mathcal{G}(t,x) = t\mathcal{K}x + (1-t)\frac{1}{\varepsilon+1}\mathcal{K}x.$$

which is continuous and uniformly continuous in $t$ for each $x$, and the mapping $\mathcal{G}(t,.)$ is a strict set contraction for each $0 \leq t \leq 1$ and it has no fixed point on $\partial\mathcal{U}$. For anny $x \in \partial\mathcal{U}$, we have

$$\begin{aligned}
\|x - \mathcal{G}(t,x)\| &= \|x - t\mathcal{K}x - (1-t)\frac{1}{\varepsilon+1}\mathcal{K}x\| \\
&\geq \|x - \frac{1}{\varepsilon+1}\mathcal{K}x\| - t\|\mathcal{K}x - \frac{1}{\varepsilon+1}\mathcal{K}x\| \\
&> \gamma - \frac{\gamma}{2} > \frac{\gamma}{2}.
\end{aligned}$$

Then, from the invariance property by homotopy of the index, our claim follows. □

**Proposition 1** ([12])**.** *Let $\mathcal{P}$ be a cone in a Banach space $\mathcal{E}$. Let also, $\mathcal{U}$ be a bounded open subset of $\mathcal{P}$ with $0 \in \mathcal{U}$. Assume that $\mathcal{T} : \Omega \subset \mathcal{P} \to \mathcal{E}$ is an expansive mapping with constant $1 < h$, $S : \overline{\mathcal{U}} \to \mathcal{E}$ is a l-set contraction with $h - 1 > l \geq 0$, and $S(\overline{\mathcal{U}}) \subset (I - \mathcal{T})(\Omega)$. If there exists a positive constatnt $\varepsilon$ such that*

$$Sx \neq \{(I - \mathcal{T})(x), \quad (I - \mathcal{T})(\lambda x)\} \ \text{for all } x \in \partial\mathcal{U} \cap \Omega \text{ and } \lambda \geq 1 + \varepsilon,$$

*then, the fixed point index*

$$i_*\left(\mathcal{T} + S, \mathcal{U} \cap \Omega, \mathcal{P}\right) = 1.$$

**Proof.** The mapping $(I - \mathcal{T})^{-1}S : \overline{\mathcal{U}} \to \mathcal{P}$ is a strict set contraction and it is readily seen that the next condition is verified

$$(I - \mathcal{T})^{-1}Sx \notin \{x, \lambda x\}, \forall x \in \partial\mathcal{U} \quad \text{and} \quad \lambda \geq 1 + \epsilon.$$

It is then followed owing to the definition of $i_*$ and Lemma 1. □

We will use the following result in order to prove existence of at least two nonnegative solutions to (1).

**Theorem 4.** *Let $\mathcal{P}$ be a cone of a Banach space $\mathcal{E}$; $\Omega$ a subset of $\mathcal{P}$ and $\mathcal{U}_1, \mathcal{U}_2$ and $\mathcal{U}_3$ three open bounded subsets of $\mathcal{P}$ such that $\overline{\mathcal{U}}_1 \subset \overline{\mathcal{U}}_2 \subset \mathcal{U}_3$ and $0 \in \mathcal{U}_1$. Assume that $\mathcal{T} : \Omega \to \mathcal{P}$ is an expansive mapping with constant $h > 1$, $S : \overline{\mathcal{U}}_3 \to \mathcal{E}$ is a k-set contraction with $0 \leq k < h - 1$ and $S(\overline{\mathcal{U}}_3) \subset (I - \mathcal{T})(\Omega)$. Suppose that $(\mathcal{U}_2 \setminus \overline{\mathcal{U}}_1) \cap \Omega \neq \emptyset$, $(\mathcal{U}_3 \setminus \overline{\mathcal{U}}_2) \cap \Omega \neq \emptyset$, and there exists $v_0 \in \mathcal{P}^*$ such that the following conditions hold:*

(i) $Sx \neq (I - \mathcal{T})(x - \lambda v_0)$, *for all $\lambda > 0$ and $x \in \partial\mathcal{U}_1 \cap (\Omega + \lambda v_0)$,*

(ii)  *There exists $0 \leq \epsilon$ such that $Sx \neq (I - \mathcal{T})(\lambda x)$, $\forall 1 + \epsilon \leq \lambda$, $x \in \partial \mathcal{U}_2$ and $\lambda x \in \Omega$,*

(iii)  *$Sx \neq (I - \mathcal{T})(x - \lambda v_0)$, $\forall 0 < \lambda$ and $x \in \partial \mathcal{U}_3 \cap (\Omega + \lambda v_0)$.*

*Then, the operator $\mathcal{T} + S$ has at least two non-zero fixed points $x_1, x_2 \in \mathcal{P}$ such that*

$$x_1 \in \partial \mathcal{U}_2 \cap \Omega \text{ and } x_2 \in (\overline{\mathcal{U}}_3 \setminus \overline{\mathcal{U}}_2) \cap \Omega$$

*or*

$$x_1 \in (\mathcal{U}_2 \setminus \mathcal{U}_1) \cap \Omega \text{ and } x_2 \in (\overline{\mathcal{U}}_3 \setminus \overline{\mathcal{U}}_2) \cap \Omega.$$

**Proof.** Let $Sx = (I - \mathcal{T})x$ for $x \in \partial \mathcal{U}_2 \cap \Omega$, then we obtain a fixed point $x_1 \in \partial \mathcal{U}_2 \cap \Omega$ of $\mathcal{T} + S$. Let $Sx \neq (I - \mathcal{T})x$, $\forall x \in \partial \mathcal{U}_2 \cap \Omega$. Let us assume that $\mathcal{T}x + Sx \neq x$ on $\partial \mathcal{U}_1 \cap \Omega$ and $x \neq \mathcal{T}x + Sx$ on $\partial \mathcal{U}_3 \cap \Omega$, otherwise the conclusion has been proved. By [14] [Proposition 2.11 and Proposition 2.16] and Proposition 1, we have

$$i_* (\mathcal{T} + S, \mathcal{U}_1 \cap \Omega, \mathcal{P}) = i_* (\mathcal{T} + S, \mathcal{U}_3 \cap \Omega, \mathcal{P}) = 0 \text{ and } i_* (\mathcal{T} + S, \mathcal{U}_2 \cap \Omega, \mathcal{P}) = 1.$$

The property of the additivity for the index yields

$$i_* (\mathcal{T} + S, (\mathcal{U}_2 \setminus \overline{\mathcal{U}}_1) \cap \Omega, \mathcal{P}) = 1 \text{ and } i_* (\mathcal{T} + S, (\mathcal{U}_3 \setminus \overline{\mathcal{U}}_2) \cap \Omega, \mathcal{P}) = -1.$$

Then, using the existence property of the index, $\mathcal{T} + S$ has at least two fixed points $x_1 \in (\mathcal{U}_2 \setminus \mathcal{U}_1) \cap \Omega$ and $x_2 \in (\overline{\mathcal{U}}_3 \setminus \overline{\mathcal{U}}_2) \cap \Omega$.  $\square$

In [15], it is proved that the problem

$$^c D_{t,0+}^{\beta} v(t) = f_1(t), \quad t \in J, \quad t \neq t_k, \quad k \in \{1, \ldots, n_1\},$$

$$\partial_t v(t_k+) = \partial_t v(t_k-) + \widetilde{I}_k(v(t_k)), \quad t_k \in (0, 1),$$

$$v(t_k+) = v(t_k-) + \widetilde{L}_k(v(t_k)), \quad t_k \in (0, 1),$$

$$v(0) = h_1(v(0)), \quad v(1) = h_2(v(1)),$$

where $f_1 \in \mathcal{C}(J), h_1, h_2 \in \mathcal{C}(\mathbb{R})$, has a solution of the form

$$v(t) = \begin{cases} c_1(t, v(t))t + h_1(v(t)) + \frac{1}{\Gamma(\beta)} \int_0^t (t - s)^{\beta - 1} f_1(s) ds, \quad t \in J_0, \\[2mm] c_1(t, v(t))t + h_1(v(t)) + \frac{1}{\Gamma(\beta)} \int_{t_k}^t (t - s)^{\beta - 1} f_1(s) ds \\[2mm] \quad + \sum_{j=1}^{k} \frac{1}{\Gamma(\beta)} \int_{t_{j-1}}^{t_j} (t_j - s)^{\beta - 1} f_1(s) ds + \sum_{j=1}^{k} (t - t_j) \widetilde{I}_j(v(t_j)) \\[2mm] \quad + \sum_{j=1}^{k} \frac{t - t_j}{\Gamma(\beta - 1)} \int_{t_{j-1}}^{t_j} (t_j - s)^{\beta - 2} f_1(s) ds + \sum_{j=1}^{k} \widetilde{L}_j(v(t_j)), \quad t \in J_k, \end{cases}$$

where

$$c_1(t, v(t)) = h_2(v(t)) - h_1(v(t)) - \sum_{j=1}^{n_1+1} \frac{1}{\Gamma(\beta)} \int_{t_{j-1}}^{t_j} (t_j - s)^{\beta-1} f_1(s) ds$$

$$- \sum_{j=1}^{n_1} \widetilde{L}_j(v(t_j)) - \sum_{j=1}^{n_1} \frac{1-t_j}{\Gamma(\beta-1)} \int_{t_{j-1}}^{t_j} (t_j - s)^{\beta-2} f_1(s) ds$$

$$- \sum_{j=1}^{n_1} (1 - t_j) I_j(v(t_j)), \quad t \in J.$$

## 4. Proof of Theorem 1

For convenience, we set $\mathcal{X} = PC^2(J, \mathcal{C}^2(\mathbb{R}^n))$. For $v = v(x,t) \in \mathcal{X}$ and $(x,t) \in J \times \mathbb{R}^n$, we define the operator

$$S_1 v = \begin{cases} -v + c(t,x,v) + h_1(x, v(x,0)) \\[2mm] + \frac{1}{\Gamma(\beta)} \int_0^t (t-s)^{\beta-1} (f(.,s,v(.,s), \partial_t v(.,s), v_x(.,s)) + \Delta_x v(.,s)) ds, \\[2mm] -v + c(t,x,v) + h_1(x, v(x,0)) \\[2mm] + \frac{1}{\Gamma(\beta)} \int_{t_k}^t (t-s)^{\beta-1} (f(.,s,v(.,s), \partial_t v(.,s), v_x(.,s)) + \Delta_x v(.,s)) ds \\[2mm] + \sum_{j=1}^{k} \frac{1}{\Gamma(\beta)} \int_{t_{j-1}}^{t_j} (t_j-s)^{\beta-1} (f(.,s,v(.,s), \partial_t v(.,s), v_x(.,s)) + \Delta_x v(.,s)) ds \\[2mm] + \sum_{j=1}^{k} (t-t_j) I_j(x, t_j, v(x, t_j)) \\[2mm] + \sum_{j=1}^{k} \frac{t-t_j}{\Gamma(\beta-1)} \int_{t_{j-1}}^{t_j} (t_j-s)^{\beta-1} (f(.,s,v(.,s), \partial_t v(.,s), v_x(.,s)) + \Delta_x v(.,s)) ds \\[2mm] + \sum_{j=1}^{k} L_j(x, t_j, v(x, t_j)), \quad t \in J_k, \end{cases}$$

where

$$c(t,x,v) = h_2(x, v(x,1)) - h_1(x, v(x,0))$$

$$- \sum_{j=1}^{n_1+1} \frac{1}{\Gamma(\beta)} \int_{t_{j-1}}^{t_j} (t_j-s)^{\beta-1} (f(.,s,v(.,s), \partial_t v(.,s), v_x(.,s)) + \Delta_x v(.,s)) ds$$

$$- \sum_{j=1}^{n_1} L_j(x, t_j, v(x, t_j))$$

$$- \sum_{j=1}^{n_1} \frac{1-t_j}{\Gamma(\beta-1)} \int_{t_{j-1}}^{t_j} (t_j-s)^{\beta-1} (f(.,s,v(.,s), \partial_t v(.,s), v_x(.,s)) + \Delta_x v(.,s)) ds$$

$$- \sum_{j=1}^{n_1} (1 - t_j) I_j(x, t_j, v(x, t_j)).$$

Note that if $v \in \mathcal{X}$ satisfying

$$S_1 v = 0,$$

then $v$ is a solution to the problem (1). Set

$$
\begin{aligned}
\mathcal{B}_1 \;=\; & \mathcal{B} + 2\mathcal{B}^{1+s_1} + 2\mathcal{B}^{1+s_2} + 2\sum_{j=1}^{m-1}\left(\mathcal{B}^{1+s_{1j}} + \mathcal{B}^{1+s_{2j}}\right) \\
& + \left(\frac{n_1+3}{\Gamma(\beta+1)} + \frac{n_1+1}{\Gamma(\beta)}\right)\left(\sum_{j=1}^{r}\left(\mathcal{B}^{p_j+1} + \mathcal{B}^{q_j+1} + \sum_{i=1}^{n}\mathcal{B}^{r_{ji}+1}\right) + n\mathcal{B}\right).
\end{aligned}
$$

**Lemma 2.** *Let* $(Hyp1)$–$(Hyp4)$ *hold. For* $v \in \mathcal{X}$, $\|v\| \leq \mathcal{B}$, *we have*

$$|S_1 v| \leq \mathcal{B}_1.$$

**Proof.** We have

$$
\begin{aligned}
|\Delta_x v| \;=\; & \left|\sum_{j=1}^{n} v_{x_j x_j}\right| \\
\leq\; & \sum_{j=1}^{n} |v_{x_j x_j}| \\
\leq\; & n\mathcal{B},
\end{aligned}
$$

and

$$
\begin{aligned}
|f(t,x,v,\partial_t v, v_x)| \;\leq\; & \sum_{j=1}^{n}\left(a_j(x,t)|v|^{p_j} + b_j(x,t)|v|^{q_j}\right. \\
& \left. + \sum_{i=1}^{n} c_{ji}(x,t)|v_{x_i}|^{r_{ji}}\right) \\
\leq\; & \sum_{j=1}^{r}\left(\mathcal{B}^{p_j+1} + \mathcal{B}^{q_j+1} + \sum_{i=1}^{n}\mathcal{B}^{r_{ji}+1}\right),
\end{aligned}
$$

and

$$
\begin{aligned}
|I_k(x,t_k,v(x,t_k))| \;\leq\; & a_{1k}(x,t_k)|v(x,t_k)|^{s_{1k}} \\
\leq\; & \mathcal{B}^{1+s_{1k}},
\end{aligned}
$$

and

$$
\begin{aligned}
|L_k(x,t_k,v(x,t_k))| \;\leq\; & a_{2k}(x,t_k)|v(x,t_k)|^{s_{2k}} \\
\leq\; & \mathcal{B}^{1+s_{2k}},
\end{aligned}
$$

and

$$
\begin{aligned}
|h_1(x,v(x,0))| \;\leq\; & b_{11}(x)|v(x,0)|^{s_1} \\
\leq\; & \mathcal{B}^{1+s_1},
\end{aligned}
$$

and

$$|h_2(x, v(x, 1))| \quad \leq \quad b_{12}(x)|v(x, 1)|^{s_2}$$

$$\leq \quad \mathcal{B}^{1+s_2},$$

and

$$
\begin{aligned}
|c(t, x, v)| \quad = \quad & \Bigg| h_2(x, v(x, 1)) - h_1(x, v(x, 0)) \\
& - \sum_{j=1}^{n_1+1} \frac{1}{\Gamma(\beta)} \int_{t_{j-1}}^{t_j} (t_j - s)^{\beta-1} (f(., s, v(., s), \partial_t v(., s), v_x(., s)) + \Delta_x v(., s)) ds \\
& - \sum_{j=1}^{n_1} L_j(x, t_j, v(x, t_j)) \\
& - \sum_{j=1}^{n_1} \frac{1 - t_j}{\Gamma(\beta - 1)} \int_{t_{j-1}}^{t_j} (t_j - s)^{\beta-1} (f(., s, v(., s), \partial_t v(., s), v_x(., s)) + \Delta_x v(., s)) ds \\
& - \sum_{j=1}^{n_1} (1 - t_j) I_j(x, t_j, v(x, t_j)) \Bigg| \\[2mm]
\leq \quad & |h_2(x, v(x, 1))| + |h_1(x, v(x, 0))| \\
& + \sum_{j=1}^{n_1+1} \frac{1}{\Gamma(\beta)} \int_{t_{j-1}}^{t_j} (t_j - s)^{\beta-1} (|f(., s, v(., s), \partial_t v(., s), v_x(., s))| + |\Delta_x v(., s)|) ds \\
& + \sum_{j=1}^{n_1} |L_j(x, t_j, v(x, t_j))| \\
& + \sum_{j=1}^{n_1} \frac{1 - t_j}{\Gamma(\beta - 1)} \int_{t_{j-1}}^{t_j} (t_j - s)^{\beta-1} (|f(., s, v(., s), \partial_t v(., s), v_x(., s))| + |\Delta_x v(., s)|) ds \\
& + \sum_{j=1}^{n_1} (1 - t_j) |I_j(x, t_j, v(x, t_j))| \\[2mm]
\leq \quad & \mathcal{B}^{1+s_1} + \mathcal{B}^{1+s_2} + \sum_{j=1}^{m-1} \left( \mathcal{B}^{1+s_{1j}} + \mathcal{B}^{1+s_{2j}} \right) \\
& + \left( \frac{n_1 + 1}{\Gamma(\beta + 1)} + \frac{n_1}{\Gamma(\beta)} \right) \left( \sum_{j=1}^{r} \left( \mathcal{B}^{p_j+1} + \mathcal{B}^{q_j+1} + \sum_{i=1}^{n} \mathcal{B}^{r_{ji}+1} \right) + n\mathcal{B} \right).
\end{aligned}
$$

Hence,

$$
\begin{aligned}
|S_1 v| \;=\; & \left| -v + c(t,x,v) + h_1(x,v(x,0)) \right. \\[4pt]
& \left. + \frac{1}{\Gamma(\beta)} \int_0^t (t-s)^{\beta-1} (f(.,s,v(.,s),\partial_t v(.,s),v_x(.,s)) + \Delta_x v(.,s)) ds \right| \\[8pt]
\leq \; & |v| + |c(t,x,v)| + |h_1(x,v(x,0))| \\[4pt]
& + \frac{1}{\Gamma(\beta)} \int_0^t (t-s)^{\beta-1} (|f(.,s,v(.,s),\partial_t v(.,s),v_x(.,s))| + |\Delta_x v(.,s)|) ds \\[8pt]
\leq \; & \mathcal{B} + \left( \frac{n_1+1}{\Gamma(\beta+1)} + \frac{n_1}{\Gamma(\beta)} \right) \left( \sum_{j=1}^r \left( \mathcal{B}^{p_j+1} + \mathcal{B}^{q_j+1} + \sum_{i=1}^n \mathcal{B}^{r_{ji}+1} \right) + n\mathcal{B} \right) \\[8pt]
& + \mathcal{B}^{1+s_1} + \mathcal{B}^{1+s_2} + \sum_{j=1}^{n_1} \left( \mathcal{B}^{1+s_{1j}} + \mathcal{B}^{1+s_{2j}} \right) + \mathcal{B}^{1+s_1} \\[8pt]
& + \frac{1}{\Gamma(\beta+1)} \left( \sum_{j=1}^r \left( \mathcal{B}^{p_j+1} + \mathcal{B}^{q_j+1} + \sum_{i=1}^n \mathcal{B}^{r_{ji}+1} \right) + n\mathcal{B} \right) \\[8pt]
= \; & \mathcal{B} + 2\mathcal{B}^{1+s_1} + \mathcal{B}^{1+s_2} + \sum_{j=1}^{m-1} \left( \mathcal{B}^{1+s_{1j}} + \mathcal{B}^{1+s_{2j}} \right) \\[8pt]
& + \left( \frac{n_1+2}{\Gamma(\beta+1)} + \frac{n_1}{\Gamma(\beta)} \right) \left( \sum_{j=1}^r \left( \mathcal{B}^{p_j+1} + \mathcal{B}^{q_j+1} + \sum_{i=1}^n \mathcal{B}^{r_{ji}+1} \right) + n\mathcal{B} \right) \\[8pt]
\leq \; & \mathcal{B}_1,
\end{aligned}
$$

and

$$
\begin{aligned}
|S_1 v| \;=\; & \left| -v + c(t,x,v) + h_1(x,v(x,0)) \right. \\[4pt]
& + \frac{1}{\Gamma(\beta)} \int_{t_k}^t (t-s)^{\beta-1} (f(.,s,v(.,s),\partial_t v(.,s),v_x(.,s)) + \Delta_x v(.,s)) ds \\[8pt]
& + \sum_{j=1}^k \frac{1}{\Gamma(\beta)} \int_{t_{j-1}}^{t_j} (t_j-s)^{\beta-1} (f(.,s,v(.,s),\partial_t v(.,s),v_x(.,s)) + \Delta_x v(.,s)) ds \\[8pt]
& + \sum_{j=1}^k (t-t_j) I_j(x,t_j,v(x,t_j)) \\[8pt]
& + \sum_{j=1}^k \frac{t-t_j}{\Gamma(\beta-1)} \int_{t_{j-1}}^{t_j} (t_j-s)^{\beta-1} (f(.,s,v(.,s),\partial_t v(.,s),v_x(.,s)) + \Delta_x v(.,s)) ds \\[8pt]
& \left. + \sum_{j=1}^k L_j(x,t_j,v(x,t_j)) \right|
\end{aligned}
$$

$$\leq \quad |v| + |c(t, x, v)| + |h_1(x, v(x, 0))|$$

$$+ \frac{1}{\Gamma(\beta)} \int_{t_k}^{t} (t-s)^{\beta-1} (|f(., s, v(., s), \partial_t v(., s), v_x(., s))| + |\Delta_x v(., s)|) ds$$

$$+ \sum_{j=1}^{k} \frac{1}{\Gamma(\beta)} \int_{t_{j-1}}^{t_j} (t_j - s)^{\beta-1} (|f(., s, v(., s), \partial_t v(., s), v_x(., s))| + |\Delta_x v(., s)|) ds$$

$$+ \sum_{j=1}^{k} (t - t_j) |I_j(x, t_j, v(x, t_j))|$$

$$+ \sum_{j=1}^{k} \frac{t - t_j}{\Gamma(\beta - 1)} \int_{t_{j-1}}^{t_j} (t_j - s)^{\beta-1} (|f(., s, v(., s), \partial_t v(., s), v_x(., s))| + |\Delta_x v(., s)|) ds$$

$$+ \sum_{j=1}^{k} |L_j(x, t_j, v(x, t_j))|$$

$$\leq \quad \mathcal{B} + 2\mathcal{B}^{1+s_1} + \mathcal{B}^{1+s_2} + \sum_{j=1}^{m-1} \left( \mathcal{B}^{1+s_{1j}} + \mathcal{B}^{1+s_{2j}} \right)$$

$$+ \left( \frac{n_1 + 2}{\Gamma(\beta + 1)} + \frac{n_1}{\Gamma(\beta)} \right) \left( \sum_{j=1}^{r} \left( \mathcal{B}^{p_j+1} + \mathcal{B}^{q_j+1} + \sum_{i=1}^{n} \mathcal{B}^{r_{ji}+1} \right) + n\mathcal{B} \right)$$

$$+ \frac{1}{\Gamma(\beta + 1)} \left( \sum_{j=1}^{r} \left( \mathcal{B}^{p_j+1} + \mathcal{B}^{q_j+1} + \sum_{i=1}^{n} \mathcal{B}^{r_{ji}+1} \right) + n\mathcal{B} \right)$$

$$+ \frac{1}{\Gamma(\beta)} \left( \sum_{j=1}^{r} \left( \mathcal{B}^{p_j+1} + \mathcal{B}^{q_j+1} + \sum_{i=1}^{n} \mathcal{B}^{r_{ji}+1} \right) + n\mathcal{B} \right)$$

$$+ \sum_{j=1}^{k} \mathcal{B}B^{1+s_{1j}} + \sum_{j=1}^{k} \mathcal{B}^{1+s_{2j}}$$

$$= \quad \mathcal{B} + 2v^{1+s_1} + 2\mathcal{B}^{1+s_2} + 2 \sum_{j=1}^{m-1} \left( \mathcal{B}^{1+s_{1j}} + \mathcal{B}^{1+s_{2j}} \right)$$

$$+ \left( \frac{n_1 + 3}{\Gamma(\beta + 1)} + \frac{n_1 + 1}{\Gamma(\beta)} \right) \left( \sum_{j=1}^{r} \left( \mathcal{B}^{p_j+1} + \mathcal{B}^{q_j+1} + \sum_{i=1}^{n} \mathcal{B}^{r_{ji}+1} \right) + n\mathcal{B} \right)$$

$$= \quad \mathcal{B}_1.$$

This is completes the proof. □

Let us suppose that $\mathcal{A} \in \mathbb{R}_*^+$ and $g$ to be continuous function on $\mathbb{R}^n$, where

**(Hyp5)** $g > 0$ on $\mathbb{R}^n \setminus \{\bigcup_{i=1}^{n} \{x_i = 0\}\}$,

$$g(0, x_2, \ldots, x_n) = \ldots = g(x_1, \ldots, x_{n-1}, 0) = 0, \quad x_j \in \mathbb{R}, \quad j \in \{1, \ldots, n\},$$

and

$$2 \cdot 8^n \prod_{j=1}^{n} \left( 1 + |x_j| + x_j^2 \right) \left| \int_0^x g(y) dy \right| \leq \mathcal{A},$$

where

$$\int_0^x = \int_0^{x_1} \cdots \int_0^{x_n}, \quad dy = dy_n \ldots dy_1.$$

We define for $v \in \mathcal{X}$, the operator

$$S_2 v = \int_0^t (t-s)^2 \int_0^x \prod_{j=1}^n (x_j - y_j)^2 g(y) S_1 v(s, y) \, dy \, ds.$$

**Lemma 3.** *Suppose* $(Hyp1)$–$(Hyp5)$. *If* $v \in \mathcal{X}$ *satisfying*

$$S_2 v = 0, \tag{4}$$

*then* $v$ *satisfies the problem* (1).

**Proof.** Differentiating three times in $t$ and three times in $x_1$, ..., $x_n$ the equation (4), we obtain

$$g(x) S_1 v = 0, \quad (x,t) \in J \times \left( \mathbb{R}^n \backslash \left\{ \bigcup_{i=1}^n \{x_i = 0\} \right\} \right),$$

whereupon

$$S_1 v = 0, \quad (x,t) \in J \times \left( \mathbb{R}^n \backslash \left\{ \bigcup_{i=1}^n \{x_i = 0\} \right\} \right).$$

Since $S_1 v \in \mathcal{C}(J \times \mathbb{R}^n)$, we have

$$
\begin{aligned}
0 &= S_1 v(t, 0, x_2, \ldots, x_n) \\[2mm]
&= \lim_{x_1 \to 0} S_1 v(t, x_1, x_2, \ldots, x_n), \\[2mm]
&\quad \cdots \\[2mm]
0 &= S_1 v(t, x_1, x_2, \ldots, 0) \\[2mm]
&= \lim_{x_n \to 0} S_1 v(t, x_1, x_2, \ldots, x_n), \quad x_1, \ldots, x_n \in \mathbb{R}, \quad t \in J.
\end{aligned}
$$

Therefore, we obtain

$$S_1 v = 0.$$

Hence, we then conclude that $v$ satisfies (1). The proof is now completed. $\square$

**Lemma 4.** *Let assumptions* $(Hyp1)$–$(Hyp5)$ *hold. If* $v \in \mathcal{X}$ *and* $\|v\| \le B$, *then*

$$\|S_2 v\| \le AB_1.$$

**Proof.** We have

$$
\begin{aligned}
|S_2 v| &= \left| \int_0^t \int_0^x \prod_{j=1}^n (t-s)^2 (x_j - s_j)^2 g(t_1, s) S_1 v(t_1, s)\,ds\,dt_1 \right| \\
&\leq \int_0^t \left| \int_0^x \prod_{j=1}^n (t-s)^2 (x_j - s_j)^2 g(t_1, s) |S_1 v(t_1, s)|\,ds \right| dt_1 \\
&\leq \mathcal{B}_1 \int_0^t \left| \int_0^x \prod_{j=1}^n (x_j - s_j)^2 g(t_1, s)\,ds \right| dt_1 \\
&\leq \mathcal{B}_1 4^n \prod_{j=1}^n x_j^2 \int_0^t \left| \int_0^x g(t_1, s)\,ds \right| dt_1 \\
&\leq 2\mathcal{B}_1 8^n \prod_{j=1}^n \left( 1 + |x_j| + x_j^2 \right) \int_0^t \left| \int_0^x g(t_1, s)\,ds \right| dt_1 \\
&\leq \mathcal{A}\mathcal{B}_1,
\end{aligned}
$$

and

$$
\begin{aligned}
|\partial_t S_2 v| &= \left| 2 \int_0^t \int_0^x \prod_{j=1}^n (t-s)(x_j - s_j)^2 g(t_1, s) S_1 v(t_1, s)\,ds\,dt_1 \right| \\
&\leq 2 \int_0^t \left| \int_0^x \prod_{j=1}^n (t-s)(x_j - s_j)^2 g(t_1, s) |S_1 v(t_1, s)|\,ds \right| dt_1 \\
&\leq 2\mathcal{B}_1 \int_0^t \left| \int_0^x \prod_{j=1}^n (x_j - s_j)^2 g(t_1, s)\,ds \right| dt_1 \\
&\leq 2\mathcal{B}_1 4^n \prod_{j=1}^n x_j^2 \int_0^t \left| \int_0^x g(t_1, s)\,ds \right| dt_1 \\
&\leq 2\mathcal{B}_1 8^n \prod_{j=1}^n \left( 1 + |x_j| + x_j^2 \right) \int_0^t \left| \int_0^x g(t_1, s)\,ds \right| dt_1 \\
&\leq \mathcal{A}\mathcal{B}_1,
\end{aligned}
$$

$$
\begin{aligned}
|\partial_{tt} S_2 v| &= \left| 2 \int_0^t \int_0^x \prod_{j=1}^n (x_j - s_j)^2 g(t_1, s) S_1 v(t_1, s)\,ds\,dt_1 \right| \\
&\leq 2 \int_0^t \left| \int_0^x \prod_{j=1}^n (x_j - s_j)^2 g(t_1, s) |S_1 v(t_1, s)|\,ds \right| dt_1 \\
&\leq 2\mathcal{B}_1 \int_0^t \left| \int_0^x \prod_{j=1}^n (x_j - s_j)^2 g(t_1, s)\,ds \right| dt_1 \\
&\leq 2\mathcal{B}_1 4^n \prod_{j=1}^n x_j^2 \int_0^t \left| \int_0^x g(t_1, s)\,ds \right| dt_1 \\
&\leq 2\mathcal{B}_1 8^n \prod_{j=1}^n \left( 1 + |x_j| + x_j^2 \right) \int_0^t \left| \int_0^x g(t_1, s)\,ds \right| dt_1 \\
&\leq \mathcal{A}\mathcal{B}_1,
\end{aligned}
$$

and

$$
\begin{aligned}
|\partial x_k S_2 v| &= 2\left| \int_0^t \int_0^x \prod_{j=1,j\neq k}^n (t-s)^2 (x_j - s_j)^2 (x_k - s_k) g(t_1,s) S_1 v(t_1,s) ds dt_1 \right| \\
&\leq 2\int_0^t \left| \int_0^x \prod_{j=1,j\neq k}^n (t-s)^2 (x_j - s_j)^2 |x_k - s_k| g(t_1,s) |S_1 v(t_1,s)| ds \right| dt_1 \\
&\leq 2\mathcal{B}_1 \int_0^t \left| \int_0^x \prod_{j=1,j\neq k}^n (x_j - s_j)^2 |x_k - s_k| g(t_1,s) ds \right| dt_1 \\
&\leq \mathcal{B}_1 4^n \prod_{j=1}^n x_j^2 |x_k| \int_0^t \left| \int_0^x g(t_1,s) ds \right| dt_1 \\
&\leq \mathcal{B}_1 8^n \prod_{j=1}^n \left(1 + |x_j| + x_j^2\right) \int_0^t \left| \int_0^x g(t_1,s) ds \right| dt_1 \\
&\leq \mathcal{A}\mathcal{B}_1, \quad k \in \{1,\ldots,n\},
\end{aligned}
$$

and

$$
\begin{aligned}
|\partial x_{kk} S_2 v| &= 2\left| \int_0^t \int_0^x \prod_{j=1,j\neq k}^n (t-s)^2 (x_j - s_j)^2 g(t_1,s) S_1 v(t_1,s) ds dt_1 \right| \\
&\leq 2\int_0^t \left| \int_0^x \prod_{j=1,j\neq k}^n (t-s)^2 (x_j - s_j)^2 g(t_1,s) |S_1 v(t_1,s)| ds \right| dt_1 \\
&\leq 2\mathcal{B}_1 \int_0^t \left| \int_0^x \prod_{j=1,j\neq k}^n (x_j - s_j)^2 g(t_1,s) ds \right| dt_1 \\
&\leq \mathcal{B}_1 4^{n-1} \prod_{j=1}^n x_j^2 \int_0^t \left| \int_0^x g(t_1,s) ds \right| dt_1 \\
&\leq \mathcal{B}_1 8^n \prod_{j=1}^n \left(1 + |x_j| + x_j^2\right) \int_0^t \left| \int_0^x g(t_1,s) ds \right| dt_1 \\
&\leq \mathcal{A}\mathcal{B}_1, \quad k \in \{1,\ldots,n\}.
\end{aligned}
$$

Thus,
$$
\|S_2 u\| \leq \mathcal{A}\mathcal{B}_1.
$$

The proof is now completed. □

Moreover, we suppose that

**(Hyp6)** $\epsilon \in (0,1)$, $\mathcal{A}$ and $\mathcal{B}$ satisfy $\epsilon \mathcal{B}_1 (1 + \mathcal{A}) < 1$ and $\mathcal{A}\mathcal{B}_1 < 1$.

Let $\widetilde{\widetilde{\widetilde{\omega}}}$ denote the set of all equi-continuous families in $\mathcal{X}$ with respect to the norm $\|\cdot\|$. Let also, $\widetilde{\widetilde{\omega}} = \overline{\widetilde{\widetilde{\widetilde{\omega}}}}$ be the closure of $\widetilde{\widetilde{\widetilde{\omega}}}$, $\widetilde{\omega} = \widetilde{\widetilde{\omega}} \cup \{h_1, h_2\}$,

$$
\omega = \{v \in \widetilde{\omega} : \|v\| \leq \mathcal{B}\}.
$$

Note that $\omega$ is a compact set in $\mathcal{X}$. For $v \in \mathcal{X}$, we define

$$
\mathcal{T}v = -\epsilon v,
$$

$$
Sv = v + \epsilon v + \epsilon S_2 v.
$$

For $v \in \varpi$, by Lemma 4, we obtain

$$
\begin{aligned}
\|(I - S)v\| &= \|\epsilon v - \epsilon S_2 v\| \\[2mm]
&\leq \epsilon \|v\| + \epsilon \|S_2 v\| \\[2mm]
&\leq \epsilon \mathcal{B}_1 + \epsilon v \mathcal{B}_1 \\[2mm]
&= \epsilon \mathcal{B}_1 (1 + \mathcal{A}) \\[2mm]
&< \mathcal{B}.
\end{aligned}
$$

Thus, $S : \varpi \to \mathcal{E}$ is continuous and $(I - S)(\varpi)$ resides in a compact subset of $\mathcal{E}$. Now, suppose that there is a $v \in \mathcal{E}$ so that $\|v\| = \mathcal{B}$ and

$$
v = \lambda(I - S)v,
$$

or

$$
\frac{1}{\lambda} v = (I - S)v = -\epsilon v - \epsilon S_2 v,
$$

or

$$
\left( \frac{1}{\lambda} + \epsilon \right) v = -\epsilon S_2 v,
$$

for some $\lambda \in \left( 0, \frac{1}{\epsilon} \right)$. Hence, $\|S_2 v\| \leq \mathcal{A} \mathcal{B}_1 < \mathcal{B}$,

$$
\epsilon \mathcal{B} < \left( \frac{1}{\lambda} + \epsilon \right) \mathcal{B} = \left( \frac{1}{\lambda} + \epsilon \right) \|v\| = \epsilon \|S_2 v\| < \epsilon \mathcal{B},
$$

which is a contradiction. Hence by Theorem 3, it follows that the operator $\mathcal{T} + S$ has a fixed point $v^* \in \varpi$. Therefore

$$
\begin{aligned}
v^* &= \mathcal{T} v^* + S v^* \\[2mm]
&= -\epsilon v^* + v^* + \epsilon v^* + \epsilon S_2 v^*,
\end{aligned}
$$

whereupon

$$
0 = S_2 v^*.
$$

Owing to the Lemma 3, we can easily conclude that $v$ is a solution to (1), which completes the proof.

## 5. Proof of the Second Result: Theorem 2

In this section, we suppose the following additional condition.

**(Hyp7)** Let $0 < m$ be a large enough and $\mathcal{B}, \mathcal{A}, L, r, R_1$ be positive constants such that

$$
R_1 > L > r, \quad 0 < \epsilon, \quad \left( \frac{2}{5m} + 1 \right) L < R,
$$

$$
\mathcal{A} \mathcal{B}_1 < \frac{L}{5}.
$$

Let

$$
\widetilde{P} = \{ v \in \mathcal{X} : v \geq 0 \quad \text{on} \quad J \times \mathbb{R}^n \}.
$$

With $\mathcal{P}$ we will denote the set of all equi-continuous families in $\widetilde{P}$. For $v \in \mathcal{X}$, define the operators

$$\mathcal{T}_1 v(t) = (1 + m\epsilon)v(t) - \epsilon \frac{L}{10},$$

$$S_3 v(t) = -\epsilon S_2 v(t) - m\epsilon v(t) - \epsilon \frac{L}{10},$$

$t \in [0, \infty)$. Note that any fixed point $v \in \mathcal{X}$ of the operator $\mathcal{T}_1 + S_3$ is a solution to the BVP (1). Define

$$\mathcal{U}_1 = \mathcal{P}_r = \{v \in \mathcal{P} : \|v\| < r\},$$

$$\mathcal{U}_2 = \mathcal{P}_L = \{v \in \mathcal{P} : \|v\| < L\},$$

$$\mathcal{U}_3 = \mathcal{P}_{R_1} = \{v \in \mathcal{P} : \|v\| < R_1\},$$

$$R_2 = R_1 + \frac{\mathcal{A}}{m}\mathcal{B}_1 + \frac{L}{5m},$$

$$\Omega = \overline{\mathcal{P}_{R_2}} = \{v \in \mathcal{P} : \|v\| \le R_2\}.$$

1.  For $v_1, v_2 \in \Omega$, we have

    $$\|\mathcal{T}_1 v_1 - \mathcal{T}_1 v_2\| = (1 + m\varepsilon)\|v_1 - v_2\|,$$

    whereupon $\mathcal{T}_1 : \Omega \to \mathcal{X}$ be an expansive operator with the constant $1 < 1 + m\varepsilon = h$.
2.  For $v \in \overline{\mathcal{P}_{R_1}}$, we obtain

    $$\|S_3 v\| \le \varepsilon \|S_2 v\| + m\varepsilon \|v\| + \varepsilon \frac{L}{10}$$

    $$\le \varepsilon \left( \mathcal{A}\mathcal{B}_1 + mR_1 + \frac{L}{10} \right).$$

    Therefore, $S_3(\overline{\mathcal{P}_{R_1}})$ is uniformly bounded. Since $S_3 : \overline{\mathcal{P}_{R_1}} \to \mathcal{X}$ is continuous, we have that $S_3(\overline{\mathcal{P}_{R_1}})$ is equi-continuous. Consequently, $S_3 : \overline{\mathcal{P}_{R_1}} \to \mathcal{X}$ is a 0-set contraction.
3.  Let $v_1 \in \overline{\mathcal{P}_{R_1}}$. Set

    $$v_2 = v_1 + \frac{1}{m}S_2 v_1 + \frac{L}{5m}.$$

    Note that $S_2 v_1 + \frac{L}{5} \ge 0$ on $J \times \mathbb{R}^n$. We have $v_2 \ge 0$ on $J \times \mathbb{R}^n$ and

    $$\|v_2\| \le \|v_1\| + \frac{1}{m}\|S_2 v_1\| + \frac{L}{5m}$$

    $$\le R_1 + \frac{\mathcal{A}}{m}\mathcal{B}_1 + \frac{L}{5m}$$

    $$= R_2.$$

    Therefore, $v_2 \in \Omega$ and

    $$-\varepsilon m v_2 = -\varepsilon m v_1 - \varepsilon S_2 v_1 - \varepsilon \frac{L}{10} - \varepsilon \frac{L}{10}$$

or

$$(I - \mathcal{T}_1)v_2 \quad = \quad -\varepsilon m v_2 + \varepsilon \frac{L}{10}$$

$$= \quad S_3 v_1.$$

Consequently, $S_3(\overline{\mathcal{P}_{R_1}}) \subset (I - \mathcal{T}_1)(\Omega)$.

4.  Assume that $\forall v_0 \in \mathcal{P}^*$ there exist $\lambda \geq 0$ and $x \in \partial \mathcal{P}_r \cap (\Omega + \lambda v_0)$ or $x \in \partial \mathcal{P}_{R_1} \cap (\Omega + \lambda v_0)$ such that

$$S_3 x = (I - \mathcal{T}_1)(x - \lambda v_0).$$

Then

$$-\epsilon S_2 x - m \epsilon x - \epsilon \frac{L}{10} = -m \epsilon (x - \lambda v_0) + \epsilon \frac{L}{10},$$

or

$$-S_2 x = \lambda m v_0 + \frac{L}{5}.$$

Hence,

$$\|S_2 x\| = \left\| \lambda m v_0 + \frac{L}{5} \right\| > \frac{L}{5}.$$

which makes a contradiction.

5.  Suppose that $\forall \epsilon_1 \geq 0$ small enough there exist a $x_1 \in \partial \mathcal{P}_L$ and $\lambda_1 \geq 1 + \epsilon_1$ such that $\lambda_1 x_1 \in \overline{\mathcal{P}}_{R_1}$ and

$$S_3 x_1 = (I - \mathcal{T}_1)(\lambda_1 x_1). \tag{5}$$

In particular, for $\epsilon_1 > \frac{2}{5m}$, we have $x_1 \in \partial \mathcal{P}_L$, $\lambda_1 x_1 \in \overline{\mathcal{P}}_{R_1}$, $\lambda_1 \geq 1 + \epsilon_1$ and (5) holds. Since $x_1 \in \partial \mathcal{P}_L$ and $\lambda_1 x_1 \in \overline{\mathcal{P}}_{R_1}$, then

$$\left( \frac{2}{5m} + 1 \right) L < \lambda_1 L = \lambda_1 \|x_1\| \leq R_1.$$

Moreover,

$$-\epsilon S_2 x_1 - m \epsilon x_1 - \epsilon \frac{L}{10} = -\lambda_1 m \epsilon x_1 + \epsilon \frac{L}{10},$$

or

$$S_2 x_1 + \frac{L}{5} = (\lambda_1 - 1) m x_1.$$

From here,

$$2 \frac{L}{5} \geq \left\| S_2 x_1 + \frac{L}{5} \right\| = (\lambda_1 - 1) m \|x_1\| = (\lambda_1 - 1) m L$$

and

$$\frac{2}{5m} + 1 \geq \lambda_1,$$

which is a contradiction.

Therefore, all conditions of Theorem 2 hold. Hence, the BVP (1) has at least two solutions $v_1$ and $v_2$ so that

$$\|v_1\| = L < \|v_2\| < R_1,$$

or

$$r < \|v_1\| < L < \|v_2\| < R_1.$$

## 6. Illustrative Example

In this example, we try to illustrate the aim of our main results. For this end, let $m = 2$, $n = 1$,

$$s_1 = s_2 = 0, \quad s_{1k} = s_{2k} = 2, \quad k \in \{1, 2\}, \quad p_1 = 3, \quad q_1 = 0, \quad r_{11} = 0,$$

$t_1 = \frac{1}{4}$, $t_2 = \frac{1}{2}$ and

$$R_1 = \mathcal{B} = 10, \quad L = 5, \quad r = 4, \quad m = 10^{50}, \quad \mathcal{A} = \frac{1}{10\mathcal{B}_1}, \quad \epsilon = \frac{1}{5\mathcal{B}_1(1+\mathcal{A})}.$$

Then

$$\mathcal{A}\mathcal{B}_1 = \frac{1}{10} < \mathcal{B}, \quad \epsilon\mathcal{B}_1(1+\mathcal{A}) < 1,$$

i.e., $(Hyp6)$ holds, and

$$r < L < R_1, \quad \epsilon > 0, \quad R_1 > \left(\frac{2}{5m}+1\right)L, \quad \mathcal{A}\mathcal{B}_1 < \frac{L}{5}.$$

i.e., $(Hyp7)$ holds. Let

$$h(s) = \log\frac{1 + s^{11}\sqrt{2} + s^{22}}{1 - s^{11}\sqrt{2} + s^{22}}, \quad l(s) = \arctan\frac{s^{11}\sqrt{2}}{1 - s^{22}}, \quad s \in \mathbb{R}, \quad s \neq \pm 1.$$

Then

$$h'(s) = \frac{22\sqrt{2}s^{10}(1 - s^{22})}{(1 - s^{11}\sqrt{2} + s^{22})(1 + s^{11}\sqrt{2} + s^{22})},$$

$$l'(s) = \frac{11\sqrt{2}s^{10}(1 + s^{20})}{1 + s^{40}}, \quad s \in \mathbb{R}, \quad s \neq \pm 1.$$

Therefore

$$-\infty < \lim_{s \to \pm\infty}(1 + s + s^2)h(s) < \infty,$$

$$-\infty < \lim_{s \to \pm\infty}(1 + s + s^2)l(s) < \infty.$$

Hence, there exists $C_1 >$ so that

$$(1 + s + s^2)^3\left(\frac{1}{44\sqrt{2}}\log\frac{1 + s^{11}\sqrt{2} + s^{22}}{1 - s^{11}\sqrt{2} + s^{22}} + \frac{1}{22\sqrt{2}}\arctan\frac{s^{11}\sqrt{2}}{1 - s^{22}}\right) \leq C_1,$$

$s \in \mathbb{R}$. We have $\lim_{s \to \pm 1} l(s) = \frac{\pi}{2}$ and as in [16] (pp. 707, Integral 79), we have

$$\int\frac{dz}{1 + z^4} = \frac{1}{4\sqrt{2}}\log\frac{1 + z\sqrt{2} + z^2}{1 - z\sqrt{2} + z^2} + \frac{1}{2\sqrt{2}}\arctan\frac{z\sqrt{2}}{1 - z^2}.$$

Let

$$Q(s) = \frac{s^{10}}{(1 + s^{44})(1 + s + s^2)^2}, \quad s \in \mathbb{R},$$

and

$$g_1(x) = Q(x_1)\ldots Q(x_n).$$

Then, there exists a positive constant $C > 0$ such that

$$2 \cdot 8^n \prod_{j=1}^{n}\left(1 + |x_j| + x_j^2\right)\left|\int_0^x g_1(y)dy\right| \leq C.$$

Let

$$g(x) = \frac{\mathcal{A}}{C}g_1(x).$$

Then

$$2 \cdot 8^n \prod_{j=1}^{n} \left(1 + |x_j| + x_j^2\right) \left| \int_0^x g(y)dy \right| \leq \mathcal{A},$$

i.e., $(Hyp7)$ holds. Then, for $x \in \mathbb{R}$, the next problem

$$
\begin{aligned}
{}^cD_{t,0+}^{\frac{5}{3}} u - v_{xx} &= \frac{u^3}{1+x^4}, \quad t \in [0,1], \\
v(t_1^+, x) &= v(t_1, x) + \frac{(v(t_1,x))^2}{1+x^{10}}, \\
v(t_2^+, x) &= v(t_2, x) + \frac{(v(t_2,x))^2}{1+x^{18}}, \\
\partial_t v(t_1^+, x) &= \partial_t v(t_1, x) + \frac{(v(t_1,x))^2}{10+20x^{30}}, \\
\partial_t v(t_2^+, x) &= \partial_t v(t_2, x) + \frac{(v(t_2,x))^2}{1+4x^{20}}, \\
v(x, 0) &= \frac{1}{1+x^4}, \\
v(x, 1) &= \frac{1}{1+x^6},
\end{aligned}
$$

is fulfilled all conditions of Theorems 1 and 2.

## 7. Conclusions

In this paper, we investigate a class of fractional impulsive wave equation. We reduce the considered problem to a suitable integral equation. Then, we define two operators and show that any fixed point of their sum is a solution of the considered problem. After this, we apply recent fixed point theorems and we show that the considered problem has at least one and at least two classical solutions. The proposed approach can be applied for other classes impulsive partial differential equations.

**Author Contributions:** S.G.: visualization; supervision. K.B.: writing—review and editing. K.Z.: writing—original draft preparation. In addition, all authors have co-operated with each other in revising the paper. All authors have read and agreed to the published version of the manuscript.

**Funding:** This research received no external funding.

**Data Availability Statement:** Not applicable.

**Conflicts of Interest:** The authors declare no conflict of interest.

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
