# Peer review of "A New Topological Approach to Target the Existence of Solutions for Nonlinear Fractional Impulsive Wave Equations"

_axioms, doi:10.3390/axioms11120721_

Round 1
Reviewer 1 Report
In this paper, to improve their previous article results authors considered a class of fractional wave equations. To mitigate the issues related on the impulsive equations, authors employed a new topological approach and proved the existence of classical solutions with a complex argument caused by impulsive perturbations.
Authors have provided enough introduction, so it is easy to understand their research work.
Authors have clearly stated their problem in the introduction and briefly described by the hypotheses and at the end of the introduction, they stated two theorems to be proved. I am impressed by their presentation at this point.
In the preliminary section, authors stated theorems for sum of two operators which seems quite convincing to me.
All the definitions have been clearly given in an order to grasp the main idea of the article. After that, proofs of the propositions and theorems are given in a logical order. My only concern in this section is the length of the proof of some theorems which could be reduced to some extent.
Overall, paper is well written, extensive amounts of time is given on proof reading so very few errors are present, properly cited the articles, and the flow of the material is very nice.
So, I would like to recommend this article for publication in this form.
Author Response
Thank you.
Reviewer 2 Report
Please consider the attached file.

Author Response
Please, see attached file
